# Recycling the Discriminator
# for Improving the Inference Mapping of GAN

## Abstract

Generative adversarial networks (GANs) have achieved outstanding success in generating the high-quality data. Focusing on the generation process, existing GANs learn a unidirectional mapping from the latent vector to the data. Later, various studies point out that the latent space of GANs is semantically meaningful and can be utilized for advanced data analysis and manipulation. In order to analyze the real data in the latent space of GANs, it is necessary to investigate the inverse generation mapping from the data to the latent vector. To tackle this problem, the bidirectional generative models introduce an encoder to establish the inverse path of the generation process. Unfortunately, their inference mapping does not accurately predict the latent vector from the data because the imperfect generator rather interferes the encoder training. In this paper, we propose an effective algorithm to accurately infer the latent vector based on existing unidirectional GANs. It is important to note that we focus on increasing the accuracy and efficiency of the inference mapping but not influencing the GAN performance toward image quality or diversity. Instead, utilizing the proposed inference mapping algorithm, we suggest a new metric for evaluating the GAN models by measuring the reconstruction error of unseen real data. The experimental analysis demonstrates that the proposed algorithm achieves more accurate inference mapping than existing methods and provides the robust metric for evaluating GAN performance.

## 1 Introduction

Generative adversarial networks (GANs) have reported a remarkable progress for successfully reproducing the real data distribution, particularly natural images. Although GANs imposes few constraint or assumption on their model definition, even without the variational bound, it is capable of producing sharp and realistic images. To this end, training the GANs involves the adversarial competition between a generator and a discriminator; the generator learns the generation process formulated by mapping the latent distribution $P_z$ to the data distribution $P_{data}$; the discriminator evaluates the generation quality by distinguishing generated images from real images. Goodfellow et al. (2014) formulates the objective of this adversarial training using the following minimax game:

$$\min_{G} \max_{D} \mathop{\mathbb{E}}_{x \sim P_{\text{data}}} \left[ \log(D(x)) \right] + \mathop{\mathbb{E}}_{z \sim P_z} \left[ \log(1 - D(G(z))) \right] ,$$

where $\mathbb{E}$ denotes expectation, $G$ and $D$ are the generator and the discriminator respectively, and $x$ and $z$ are samples drawn from $P_z$ and $P_{data}$ respectively. Once the generator learns the mapping from the latent to the data distribution, it is possible to generate arbitrary data corresponding to randomly drawn $z$. Because the generator network never observes the real data directly during training, it does not memorize the training dataset, thus produces unseen data. Since this pioneer work, various GAN models have been developed to improve the training stability, the image quality, or the diversity of generation process.

Recent studies find that the latent space of GANs derives the semantically meaningful representation (Mikolov et al., 2013). Benefit from its semantic power, several studies (Radford et al., 2016; Berthelot et al., 2017) show that it is possible to utilize the latent space of GANs for data augmentation or image editing. To further understand and interpret the semantic representation derived by the latent space of GANs, we should investigate the inference mapping from $x$ to $z$. Learning this inference

mapping can be formulated as minimizing the reconstruction error by improving the latent estimation. CoGAN (Liu & Tuzel, 2016) and BEGAN (Berthelot et al., 2017) made attempt to solve the inverse mapping from $x$ to $z$ using the non-convex optimization. More specifically, the problem can be defined as $\min_z d(x, G(z))$, where $d(\cdot)$ is the distance metric. It is important to note that this optimization aims to find the inverse generation path $G^{-1}(x)$ using the non-convex optimizer. Because of the non-linearity and model complexity of generator, calculating the inverse path suffers from multiple local minima, thus hard to reach the global optimum. Also, it is impractical due to its computational complexity at runtime. To reduce the computational complexity and improve the accuracy of inference mapping, iGAN (Zhu et al., 2016) first suggest a hybrid method; they predict the initial latent vector of $x$ using a naïve encoder and then perform the non-convex optimization to compute the best estimate of $z$ by minimizing the pixel difference between $G(z)$ and $x$.

Several studies pay attention to simultaneously learning the inference (i.e., from $x$ to $z$) and the generation path (i.e., from $z$ to $x$). The core idea of these approaches is to employ an encoder to the GAN models for achieving the bidirectional mapping between $z$ and $x$. Dumoulin et al. (2017); Donahue et al. (2017) propose novel GAN frameworks for establishing the bidirectional mapping between $P_z$ and $P_{data}$ jointly learnt by adversarial training. Both methods can be used to infer the latent vector of real data and alleviate the mode collapse problem. However, their performance is relatively poor in terms of either generation quality or inference accuracy; the poor generation leads to blurry images and the poor inference results in inaccurate inference mapping.

Alternatively, Variational Autoencoder (VAE) (Kingma & Welling, 2013) and Adversarial Autoencoder (AAE) (Makhzani et al., 2016) are the most representative generative models that explicitly learn the bidirectional mapping between $z$ and $x$. Their model architectures are quite similar to the structure of autoencoder (Baldi, 2012), composed of an encoder (i.e., the inverse generator) and a decoder (i.e., the generator). Unlike autoencoder, VAE and AAE enforce to match the latent distributions to prior distributions, thus enabling the data generation. While VAE utilizes KL divergence to match the latent to the target distribution, AAE utilizes the adversarial learning for latent distribution matching. Although both algorithms establish the bidirectional mapping between the latent and the data distribution as well as training stability, their image quality is relatively worse than those of unidirectional GANs. Specifically, their generated images are blurry and the details are lost.

In this paper, we propose an effective algorithm to infer the latent vector based on existing unidirectional GANs by maintaining their generation quality. Our goal is to improve the accuracy and efficiency of inference mapping better than other inference mapping techniques. Our idea is motivated by the fact that the discriminator of GANs can serve as a meaningful feature extractor for both real and fake images. From this intuition, we utilize the discriminator as the feature extractor for inference mapping. By fixing the feature extractor, training our inference model is to learn the mapping from the feature to the latent vector with much fewer model parameters.

We claim that our idea of recycling discriminator's features is more powerful than building new feature extractors using encoders. Learning the new feature extractor associates either fake images or real images, but not both. Because the GAN discriminator observes both real and fake images for classification, its features should be representable for both real and fake images. As our inference mapping utilizes these discriminator features, our algorithm is equally effective to the inference of both real and fake images. Besides, the proposed inference mapping performs the feature-to-latent translation in a much lower dimensional space than the image-to-latent translation of encoder-based approaches, thus this helps reduce the number of model parameters and lead the efficient model training.

As mentioned above, the proposed method focuses on improving efficiency and accuracy for deriving inference mapping compared to existing methods. For that, we learn the inference mapping independently from baseline, unidirectional GAN training. Note that we do not aim to improve generation quality or diversity as existing GAN studies. Instead, we introduce an efficient inference mapping algorithm with two major applications: 1) manipulating the image by disentangling the latent space and 2) suggesting a new metric for assessing the GAN model by measuring reconstruction errors of real data.

## 2 RELATED WORK

Existing generative models for inducing inference mapping can be categorized into two groups. The first group utilizes the structure of an autoencoder and the other group develops the joint training scheme for learning bidirectional mapping.

### 2.1 ENCODER-DECODER ARCHITECTURE

Similar to the architecture of autoencoder, VAE (Kingma & Welling, 2013) consists of the encoder for transforming the input data to the latent vector and the decoder for reconstructing the input data from the latent representation. While the autoencoder is designed for the dimensionality reduction, VAE is capable of generating samples as well by fitting the latent distribution to the prior distribution (Wainwright et al., 2008). Because the decoder learns a mapping from the prior distribution to the data distribution during training, the trained decoder serves as the generator. The weakness of VAE is that the generated images from VAE often exhibit blurs and the lack of diversity; generated images are similar to those in the training dataset. It is because the objective of the decoder is to minimize the reconstruction errors. Nevertheless, VAE is an attractive generative model due to its efficient and stable training.

### 2.2 BIDIRECTIONAL GANS

ALI (Dumoulin et al., 2017) and BiGAN (Donahue et al., 2017) suggest a new theoretical framework for training the bidirectional GANs, that jointly learn the bidirectional mapping between $P_z$ and $P_{data}$ in an unsupervised manner. They use the generator similar to unidirectional GANs (Radford et al., 2016; Mao et al., 2017; Gulrajani et al., 2017; Warde-Farley & Bengio, 2017) for constructing the forward mapping from $P_z$ to $P_{data}$, and then use the encoder to model the inference mapping from $P_{data}$ to $P_z$. To train the generator and the encoder simultaneously, they define a new objective function for the discriminator, which distinguishes the joint distribution of $\{G(z), z\}$ from that of $\{x, E(x)\}$. Note that E and G represent the mapping functions defined by the encoder and the generator, respectively. Although their models can reconstruct the original image from the estimated latent vector, the visual quality of the generation is generally worse than the unidirectional GANs. It is because of the convergence issue as pointed by (Li et al., 2017). We conjecture that this is because the imperfect generator during training misleads the training of the encoder and vice versa. Because of both the degraded generator and the inaccurate inference, the reconstruction quality is also poor; it is not faithful to preserving the characteristics of the original image.

To overcome this practical convergence issue in joint distribution matching, VEEGAN (Srivastava et al., 2017) and ALICE (Li et al., 2017) introduce an additional constraint, namely reconstruction penalty, which enforces the reconstructed image (or latent) from the estimated latent vector (or image) to be identical to the original image (or latent). Specifically, VEEGAN utilizes the inference mapping to solve mode collapse problem. To mitigate the quality degradation due to the convergence issue, they minimize the cross-entropy between $P_z$ and $E(x)$, defined as the reconstruction penalty in the latent space. The idea of reconstruction penalty in the latent space is also introduced in infoGAN (Chen et al., 2016), but they do not aim to build the inference mapping. ALICE utilizes a conditional entropy, defined as the cycle consistency (Zhu et al., 2017), to improve the learning instability of joint distribution matching. Although both methods improve the performance of joint distribution matching, they still suffer from the discrepancy between theoretical optimum and practical convergence (Li et al., 2017). This results in either slight blurriness in image generation or inaccurate inference mapping. Again, to bypass the convergence issue, we completely separate the training inference mapping updates to both generator updates and discriminator updates.

## 3 PROPOSED ALGORITHM

We propose a simple yet powerful algorithm to induce inference mapping based on the existing unidirectional GANs. We focus on increasing the accuracy and efficiency of the inference mapping unlike recent studies for improving the GAN performance. Utilizing the proposed inference mapping, we develop a new metric for evaluating the GAN performance, accounting both the image quality and diversity, by measuring the reconstruction errors of test dataset.

To build the inference mapping, we introduce the connection network. The connection network transfers the feature extracted from discriminator (i.e., the discriminative feature) to the corresponding latent vector. Although the discriminator is typically abandoned after GAN training, we claim that the discriminator is useful for inference mapping, because it provides rich information for establishing the generation mapping. Particularly, we focus on powerful features learned to represent both real and fake data and then reuse the discriminator as the feature extractor of our connection network. We insist that the discriminator as feature extractor should be more powerful representation than other encoder based feature extractors. It is because the discriminator observes both real and fake images while existing encoder based inference mapping observes either real or fake images but not both. Furthermore, since our problem is defined as the low-to-low dimensional data translation, we effectively gain the computational efficiency. As a result, our connection network is effective in terms of improving accuracy and efficiency of inference mapping.

The objective for learning the connection network is represented as follows:

$$\min_{\text{CN}} \left| z - \text{CN}\left(D^f\left(G\left(z\right)\right)\right)\right|^2,$$

where CN is the connection network, $D^f$ indicates the discriminative feature vector extracted from the last layer of the discriminator. Note that the generator and discriminator are fixed while the connection network is updated.

### 3.1 Connection network

Our algorithm is inspired by the report from the previous study that the GAN discriminator learns the hierarchy of features so to distinguish the real and the fake images (Radford et al., 2016). Based on this report, we claim that the well-trained discriminator can serve as a well-trained feature extractor for deriving the inference mapping. In theory, GAN training is terminated when it reaches to Nash equilibrium (i.e., the generator produces realistic samples such that the discriminator no longer distinguishes the real and fake images). One might doubt that the discriminator at Nash equilibrium is no longer optimal because it cannot distinguish the fake and real images, thus not an optimal feature extractor. If the Nash equilibrium is caused by the degradation of the discriminator power, the above statement is a reasonable argument. However, the Nash equilibrium of GAN training is not the result of weak discriminator but the result of a strong generator that produces the realistic sample. Moreover, the discriminator at the Nash equilibrium is the one that leads such a strong generator. Hence, its discriminative features are sufficiently representable for both real and fake data.

Based on this analysis, we propose the connection network that learns the mapping from the discriminative feature from the discriminator to $z \sim P_z$. The generated image from $z$ is projected onto the discriminative feature space, and then this feature vector maps to the original $z$ using the connection network. It is important to understand that the correspondences between $z$ and the discriminative features are automatically determined for any random variable $z$ once training both the generator and the discriminator ends. Because we can draw the infinite number of samples from the distribution $P_z$, the training data (i.e., a set of $z$ and its discriminative feature vector pair) are also unlimited. That means, the amount of our training samples approaches to the infinity, thus our training dataset can be regarded as a superset of the real dataset. Owing to this attractive nature of our model, although the connection network is trained only with generated samples, we empirically observe that the inference mapping is successfully established from the real data to its latent vectors.

Summarizing, as an effective solution for inference mapping, our connection network is superior to existing methods in terms of accuracy and computational efficiency. We believe that the accuracy is improved because the powerful feature representation from the discriminator facilitates to learn the inference mapping. Moreover, by reusing the features from discriminator, our problem becomes to induce the mapping from the low to the low dimensional representations. In this way, we significantly reduce the model parameters as well as computational complexity during training.

### 3.2 Metric for evaluating GAN performance

Suppose that our method performs an ideal inference mapping, then the reconstruction accuracy is directly related to the generator performance. Meanwhile, the inference performance by the connection network is governed by the effectiveness of discriminator as the feature extractor. From

this tendency, we conclude that the reconstruction performance is an important indicator of the performance of the generator and the discriminator. Therefore, we propose a new metric that aims to evaluate whether the data distribution is accurately modeled by measuring the reconstruction errors of unseen test samples. The reconstruction error is increased because 1) the generator is not strong enough to produce realistic samples (i.e., image quality), or 2) the generator does not cover all modes of data distribution (i.e., image diversity). For either case, we can consider that those samples are not correctly modeled by that GANs. Consequently, our proposed metric is an effective tool for evaluating both the image quality and diversity simultaneously. To quantitatively evaluate the reconstruction performance, we employ two metrics. Given the original data and its reconstructed one, the first is the peak signal-to-noise ratio (PSNR) that measures their average of pixel difference, and the other is the structural similarity index (SSIM) that measures on their structural difference.

In this paper, we emphasize that our metric is effective in addressing the limitations of the inception score and MS-SSIM, which are the most widely used metrics for evaluating GAN performance. The inception score evaluates the quality of generated images using a pre-trained classifier. The critical, well-known issue on inception score is that poor quality images do not necessarily map to low inception scores as discussed in (Barratt & Sharma, 2018); Salimans et al. (2016) shows that adversarial examples can greatly increase the inception score although they have poor quality. The MS-SSIM is the most popular metric for measuring the diversity of generated images. However, it is not effective if the target dataset consists of very diverse images (Fedus et al., 2018) such as CIFAR10 or ImageNet. More importantly, MS-SSIM can be manipulated by simply generating random noises because it only accounts the patch difference among generated images. Summarizing, both inception score and MS-SSIM are not robust because there are alternative ways to attack those metrics, generating either adversarial examples or random patches.

Unlike existing metrics, the proposed metric rigorously measures the image quality regardless of datasets; a poor (or high) quality image always maps to a low (or high) score on any dataset. When adapting PSNR as the difference measure, the difference between the original image and that with a slight translation can be large although they are nearly identical in terms of image context. Fortunately, this issue can be resolved with SSIM. Moreover, our metric is robust against attacks because we explicitly compare the reconstruction with the ground truth. Note that various issues arose in both inception score and MS-SSIM are often caused by the absence of ground truth. Another attractive property of the proposed metric is that it is applicable to any dataset, and our score is intuitive to interpret the performance. Note that not all dataset can be evaluated by both metrics. For example, CelebA cannot be evaluated by inception score and CIFAR10 cannot be evaluated with MS-SSIM (Odena et al., 2017). Also, inception score and MS-SSIM should be compared with its ideal performance, the score of the real dataset, to understand its level of achievement. That is, it is difficult to intuitively interpret the quality from absolute numbers. On the other hand, the proposed metric has an absolute criterion; SSIM = 1, PSNR→ inf dB for ideal performance while SSIM = 0, PSNR→ 0 dB for the worst performance.

As mentioned earlier, our metric evaluates both the image quality and diversity with a single score while existing metrics focus on either of them. Because the proposed metric utilizing image reconstruction penalizes either poor quality or poor diversity, it reports high scores only if both quality and diversity are satisfied. It is clear that PSNR and SSIM measure the quality of a single sample, thus the average of many samples generalizes the image quality. Unlike the image quality, it might be less intuitive to evaluate the image diversity by image reconstruction. We claim that the image diversity can be evaluated by the average PSNR or SSIM of many samples if those samples cover all modes of the data distribution; it is achieved by constructing the test dataset by uniformly sampling the data distribution. If the baseline GANs cannot cover various modes of data distribution, the image reconstruction is generally inaccurate, leads to the low score.

## 4 EXPERIMENTAL EVALUATION

In this section, we evaluate how accurately the proposed algorithm reconstructs the original images qualitatively and quantitatively compared with a naive encoder based inference mapping, VAE and ALI/BiGAN. For the comparison, we use five different unidirectional GANs as baseline networks and apply our connection network for inference mapping. Those of baseline networks are DCGAN (Radford et al., 2016), LSGAN (Mao et al., 2017), DFM (Warde-Farley & Bengio, 2017), RFGAN

Bang & Shim (2018) and WGAN-GP (Gulrajani et al., 2017). We intend to choose those five baseline networks because all five models are significantly different in terms of loss functions or network architectures. For example, DCGAN, LSGAN, and WGAN-GP exploit different metrics while DFM adds a denoising autoencoder to the discriminator for the robustness and RFGAN implicitly regularizes the discriminator using representative features. Evaluating with a variety of unidirectional GANs, we aim to show 1) the extendibility of the proposed algorithm and 2) the effectiveness of the proposed metric using the connection network. For the fair evaluation, the architecture of all baseline networks (e.g., the number of layers, filter size, using batch normalization) borrows that of DCGAN. The connection network is composed of only two fully connected layers: 1024 full connected layer (FC) – Batch normalization (BN) – leaky rectified linear unit (Leaky ReLU) – 1024 FC – BN – Leaky ReLU – dimension of $P_z$ FC.

For experimental evaluation, we utilize three different real datasets: Fashion MNIST (Xiao et al., 2017), CIFAR10 (Krizhevsky & Hinton, 2009), and CelebA (Liu et al., 2015), all normalized between $-1$ and $1$. Note that the input dimensionality of Fashion MNIST is $(28, 28, 1)$, CIFAR10 is $(32, 32, 3)$, and CelebA is $(64, 64, 3)$.

**Evaluating the inference mapping.** We compare the proposed method with 1) naïve encoder encoder mapping, 2) iGAN (naïve encoder encoder followed by optimization in the image domain) and 3) hybrid method using the proposed method (discriminator with CN followed by optimization). Furthermore, to investigate the capability of the discriminator as the feature extractor, we directly compare our inference mapping (discriminator with CN network) with the VGG16 model (Simonyan & Zisserman, 2015) based inference mapping (pre-trained VGG16 with CN network). Note that the VGG16 model pre-trained with ImageNet is known as the good feature extractor that characterizes perceptually meaningful representation.

Fig 1 provides the reconstructed images from each method at three different iterations and the quantitative evaluation. Our algorithm successfully synthesizes the attributes in various faces, unlike the naïve encoder. As reported in iGAN, we confirm that adopting the non-convex optimization for inference mapping significantly enhances the quantitative score (i.e., SSIM and PSNR). It is because the non-convex optimization directly minimizes the pixel-wise difference between test images and reconstructed images; the goal of the non-convex optimization is nearly equivalent to the goal of PSNR. Hence, the hybrid method improves the PSNR of any baseline encoder mapping. When we replace the naïv encoder with the proposed inference mapping, its quantitative results are better than iGAN. It is because our inference mapping predicts more accurate initial latent vector.

However, these quantitative results do not exactly match with qualitative results. The quantitative results demonstrate that hybrid inference mapping is the most effective among all others. Meanwhile, the qualitative results from the hybrid methods are generally blur or have missing important components (e.g., eye glasses, mustache, gender, wrinkles, detailed hair lines, etc.). Because the hybrid inference mapping optimizes the inference mapping in the image domain (i.e., minimizing the pixel-wise difference), the inference network finally chooses the latent vector corresponding to an average-like image. Note that there exists average-like faces among many possible faces. We conjecture that, although the generator can produce sharp images, the hybrid inference mapping strategically selects average-like faces to reduce its loss function. Meanwhile, our method (also VGG16 based inference mapping) optimizes the inference mapping in the latent domain. Thus, our inference results are sharp and better preserve semantically important attributes. From examples shown in Fig 1 and Appendix Fig 6, pixel-wise loss based methods (i.e., iGAN and Hybrid+ours) fail to capture glasses, but latent vector loss based methods (i.e., ours and VGG16) reproduce the glasses. The additional qualitative and quantitative results are presented in Appendix B.

By replacing the discriminator as feature extractor by the pre-trained VGG16 network, we observe that its inference results are also as sharp and realistic as our results. However, considering the semantic similarity between the original and reconstructed image, our inference mapping can restore unique attributes (e.g., mustache, race, age, etc.) better than the VGG16 based inference mapping. Moreover, utilizing the pre-trained VGG16 require additional memory overhead while our method does not. In terms of network capacity, VGG16 has the much deeper network than the discriminator. Thus, we conclude that the proposed inference mapping is more efficient than the VGG16 based inference mapping. From these results, we confirm that recycling discriminator as a feature extractor is effective for improving inference accuracy and reducing the computational complexity.

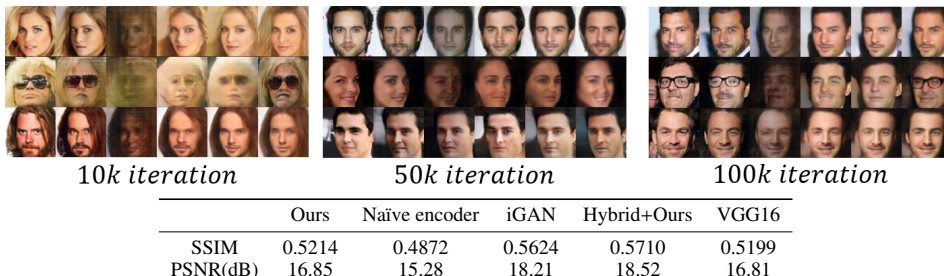

| | Ours | Naïve encoder | iGAN | Hybrid+Ours | VGG16 |
|---|---|---|---|---|---|
| SSIM | 0.5214 | 0.4872 | 0.5624 | 0.5710 | 0.5199 |
| PSNR(dB) | 16.85 | 15.28 | 18.21 | 18.52 | 16.81 |

Figure 1: Step-wise comparison. The first image is target real image and the remainders are reconstructed images by each method; the order is same with table.

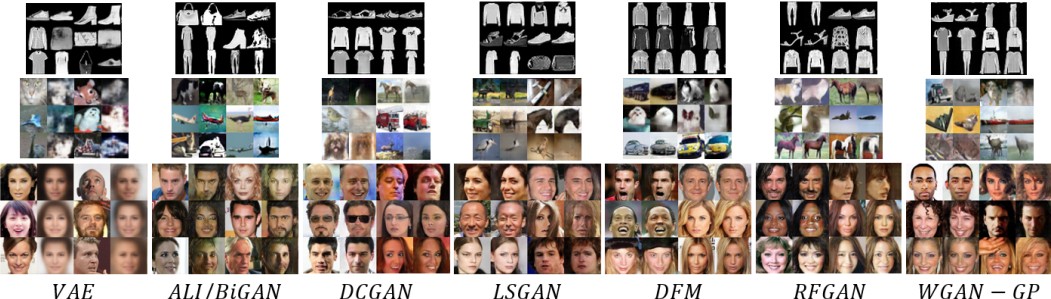

*VAE        ALI/BiGAN        DCGAN        LSGAN        DFM        RFGAN        WGAN − GP*

Figure 2: Subjective comparison for the reconstruction accuracy. The odd (first and third) columns show the input images and the even columns (second and fourth) are their corresponding reconstructed images. VAE and ALI/BiGAN are existing bidirectional generative models. We utilize five unidirectional GANs (DCGAN, LSGAN, DFM, RFGAN, and WGAN-GP) as baseline GANs and build five different variants of our models for inference mapping.

As mentioned earlier, VAE and ALI/BiGAN are representative generative models that allow inference mapping. In the following experiment, we compare our models with VAE and ALI/BiGAN. Fig 2 visualizes reconstructed images from VAE, ALI/BiGAN, and our models applied to five different baseline GANs, when the input images are at odd columns and their reconstructed images are at next even columns. The corresponding reconstruction accuracy in terms of PSNR and SSIM is summarized in Table 1. The generated images from VAE are blurry and lose the detail structures because it is optimized with a pixel reconstruction loss. Also, the less frequently appearing attributes (e.g., mustache and baldness) in the training dataset are rarely recovered in their reconstructed images. ALI/BiGAN generates sharper images than VAE. However, they are not effective to restore the important characteristics of the input image (e.g., identity), and occasionally generate completely different images from the input images. Also, from Table 1, we confirm that the accuracy of our models in both PSNR and SSIM always outperforms that of VAE and that of ALI/BiGAN.

Likewise, we observe that the reconstructed images from the variants of our models exhibit consistently better visual quality than both VAE and ALI/BiGAN. Because training our connection network focuses on the accurate inference mapping without influencing the training of the baseline GANs, the reconstructed image quality from ours is identical to that of the baseline unidirectional GANs: sharp and realistic. Furthermore, our results are superior to VAE and ALI/BiGAN in that we faithfully reconstruct the input images including various facial attributes while VAE and ALI/BiGAN often fail to handle. Based on these observations, we conclude that the proposed algorithm accurately estimates the latent vector corresponding to the input image and retains the image quality, better than other competitors.

**Experiment for image editing.**

Previously, we have shown that the latent vector corresponding to the image can be accurately estimated using our inference mapping. One of the popular applications utilizing the inference mapping is an image editing by disentangling the attributes and then modifying them for semantic editing. Previous study (Mikolov et al., 2013) reported that the simple arithmetic operation in the latent space of GANs produces a semantically meaningful combination in the representation space. Later, in DCGAN, the latent vector arithmetic is successful for facial attribute editing and produces various image effects for the first time. However, estimating the latent vector for the target image

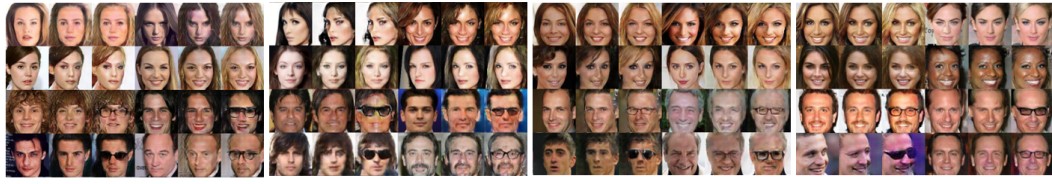

| DCGAN | LSGAN | DFM | WGAN − GP |

Figure 3: Image editing applications using latent vector arithmetic. The first and fourth column show the input images. The second and fifth column are the reconstructed images. The third and sixth column are the result of latent arithmetic. For each algorithm, the first two rows add the latent vector for blonde hairs and the last two rows add the latent vector for glasses.

Table 1: Comparison of reconstruction performance using PSNR (mean and std) and SSIM (mean and standard deviation) between test images and their reconstructed images from each model on Fashion MNIST, CIFAR10, and CelebA dataset. To verify the validity of the proposed metric, we compare our scores with the existing metrics; Inception score on CIFAR10 and MS-SSIM on CelebA. The bold indicates the top-2 results for each dataset and each metric.

| Dataset | Metric | VAE | ALI/BiGAN | DCGAN | LSGAN | DFM | RFGAN | WGAN-GP |
|---------|--------|-----|-----------|-------|-------|-----|-------|---------|
| Fashion MNIST | PSNR (dB) | 11.83 ± 0.40 | 13.02 ± 0.39 | 15.10 ± 0.38 | 15.25 ± 0.38 | 15.74 ± 0.37 | **15.94** ± 0.37 | **15.80** ± 0.36 |
| | SSIM | 0.3921 ± 0.0261 | 0.3623 ± 0.0221 | 0.5332 ± 0.0211 | **0.5665** ± 0.0203 | 0.5434 ± 0.0187 | **0.5848** ± 0.0188 | 0.5233 ± 0.0191 |
| CIFAR10 | PSNR (dB) | 13.38 ± 0.23 | 11.04 ± 0.23 | 15.60 ± 0.21 | 15.85 ± 0.22 | 16.84 ± 0.19 | **16.93** ± 0.20 | **17.08** ± 0.18 |
| | SSIM | 0.2221 ± 0.0113 | 0.1809 ± 0.0136 | 0.4250 ± 0.0101 | 0.4398 ± 0.0112 | 0.4487 ± 0.0092 | **0.4921** ± 0.0098 | **0.5061** ± 0.0089 |
| | Inception score | 5.2783 | 5.3489 | 6.4323 | 6.0821 | 6.6554 | **6.6887** | **6.6869** |
| CelebA | PSNR (dB) | 14.41 ± 0.14 | 11.94 ± 0.12 | 16.84 ± 0.26 | 16.60 ± 0.19 | 17.00 ± 0.13 | **18.60** ± 0.15 | **18.26** ± 0.22 |
| | SSIM | 0.4001 ± 0.0080 | 0.2732 ± 0.0085 | 0.5123 ± 0.0102 | 0.5611 ± 0.0096 | 0.5434 ± 0.0065 | **0.6301** ± 0.0072 | **0.6189** ± 0.0079 |
| | MS-SSIM | 0.4527 | 0.3929 | 0.4432 | **0.3901** | 0.4213 | 0.4002 | **0.3837** |

is not trivial. Even utilizing the state-of-the-art non-convex optimization, it often falls into local minima and computationally intractable for real applications. The proposed method is attractive in that we achieve the accurate latent estimation in real time (i.e., forward propagation). That means, our algorithm can be used to edit image attributes in real time. We demonstrate the results from the variants of our models as shown in Fig 3. From this figure, the first to the second row are the result of adding the mean latent vector for blonde hair with the female face, and the third to fourth column are the result of adding the mean latent vector for glasses with the male face; the images are arranged in the order of test image, reconstructed images, and vector arithmetic edited image. In all cases, our algorithm successfully synthesizes the attributes in various faces.

**Experimental comparison for GAN metric.** As mentioned in Sec 3.2, we measure the reconstruction accuracy using PSNR and SSIM on test dataset and use it as a new quantitative metric for evaluating the GAN performance. Similar to the previous experiment, VAE and ALI/BiGAN are compared with five variants of our algorithms. For the fair evaluation, we use a total of 1k test images for the experiment and then report their mean and standard deviation in Table 1, both in terms of PSNR and SSIM repeated five times for each model and each dataset. We choose 1k for test samples because the reconstruction error does not change much by increasing the test samples, varying from 100 to 10k. While inception score and MS-SSIM are meaningful on a specific dataset, the proposed metric is applicable for any type of dataset. Note that we evaluate various GANs using three different resolution and different channel dataset (i.e., Fashion MNIST, CIFAR10, and CelebA) and show the consistent results, having similar ranks across different datasets.

Comparing the existing metrics on their target dataset (i.e., inception score on CIFAR10 and MS-SSIM on CelebA), we observe that the proposed metric reports similar tendency; top-2 results from ours and existing metrics significantly overlap. By closely analyzing each metric, our metric differs from others in that we account both image quality and diversity simultaneously. For example, our metric reports the worst score for ALI/BiGAN, which has relatively a good score (top-3) in MS-SSIM. This is reasonable because ALI/BiGAN utilizes the bidirectional mapping to improve the diversity but sacrificing the image quality. Since MS-SSIM only accounts the diversity, ALI/BiGAN

achieves the meaningful performance with MS-SSIM. However, our metric measures the overall performance, thereby ALI/BiGAN is penalized due to its quality degradation. Because our metric explicitly provides the absolute numbers comparable with different datasets, we can approximate the difficulty of different datasets; most GANs achieve relatively higher performance on CelebA than others. This is because CelebA includes roughly aligned images with less diversity than other datasets.

## 5 CONCLUSION

In this study, we propose a new algorithm for establishing the accurate and efficient inference mapping on existing unidirectional GANs, without sacrificing their image quality. For that, we introduce the connection network that transfers the discriminative features extracted from the last layer of the discriminator to the latent vector. Since the discriminator learns to distinguish between real and fake samples, its features should be representable for both real and fake images. Hence, we expect that recycling the discriminator's features for building inference mapping leads to more accurate inference mapping than existing methods; existing methods adopt a new feature extractor associated either fake images or real images, but not both. Moreover, because learning the transfer from feature to latent vector is modeled by a low-to-low dimensional mapping, our algorithm is computationally efficient for training.

In this paper, we introduce two major applications using the efficient inference mapping: 1) manipulating the image by latent vector arithmetic and 2) suggesting a new metric for evaluating GAN performance by measuring the reconstruction error using unseen real data. Especially, we empirically show that our metric explicitly measures GAN performance and is applicable for various datasets with the different resolution/channel while existing metrics are limited to specific datasets (i.e., Inception score and MS-SSIM). We expect that the proposed algorithm can serve the important basis of future studies for analyzing and evaluating GAN models.

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

APPENDIX

A. Application

Toward the advanced image editing application, we develop another a novel framework for simultaneously generating and manipulating the face images with desired attributes. This work utilizes the connection network proposed in this paper and is under review for the publication. The goal of this work is to develop a single unified model that can simultaneously create and edit high-quality face images with desired attributes. The proposed framework decomposes the image into the latent and attribute vector in low dimensional representation through the connection network, and then utilizes the GAN framework for mapping the low dimensional representation to the image. Fig. 4 shows the architecture of this model. In this way, this new model can handle both the generation and editing problem by learning the generator.

Fig. 5 shows the results of attribute editing from the VAE/GAN (Larsen et al., 2016), the modified cGAN (Mirza & Osindero, 2014), IcGAN (Perarnau et al., 2016), and the new model using the connection network. This new model achieves the competitive performance with the state-of-the-art attribute editing technique in terms of reconstruction and attributes editing quality.

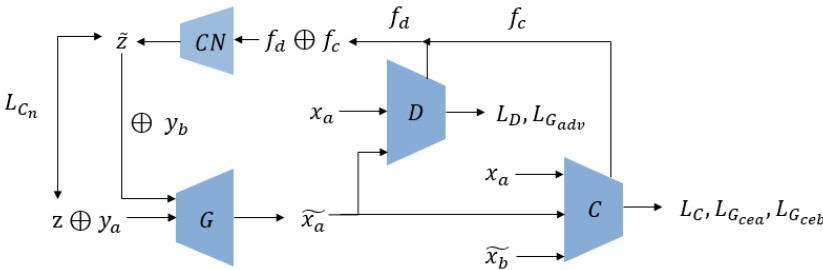

Figure 4: The network architecture of the proposed model consisting of a generator $G$, a discriminator $D$, an attribute classifier $C$, and a connection network $CN$. $\oplus$ means concatenation along the last dimension. $\tilde{x}$ and $\tilde{z}$ mean the generated image and the generated latent vector, respectively.

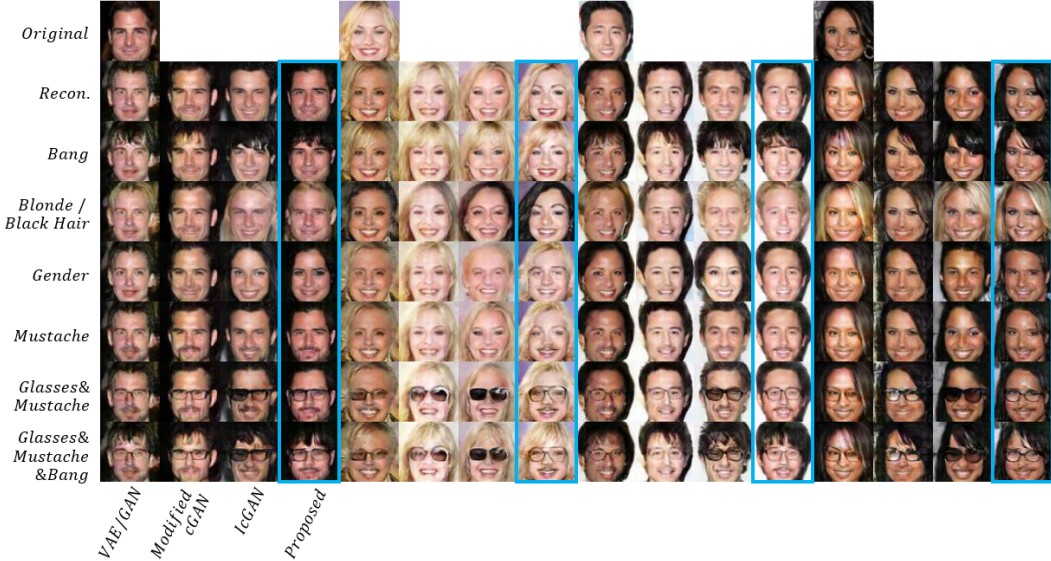

Figure 5: Comparisons of facial attribute editing. The blue box highlights our results. The first three columns are VAE/GAN, modified cGAN, and IcGAN. For each row, the specified attribute(s) is added to the input image.

B. Qualitative comparison for reconstruction performance

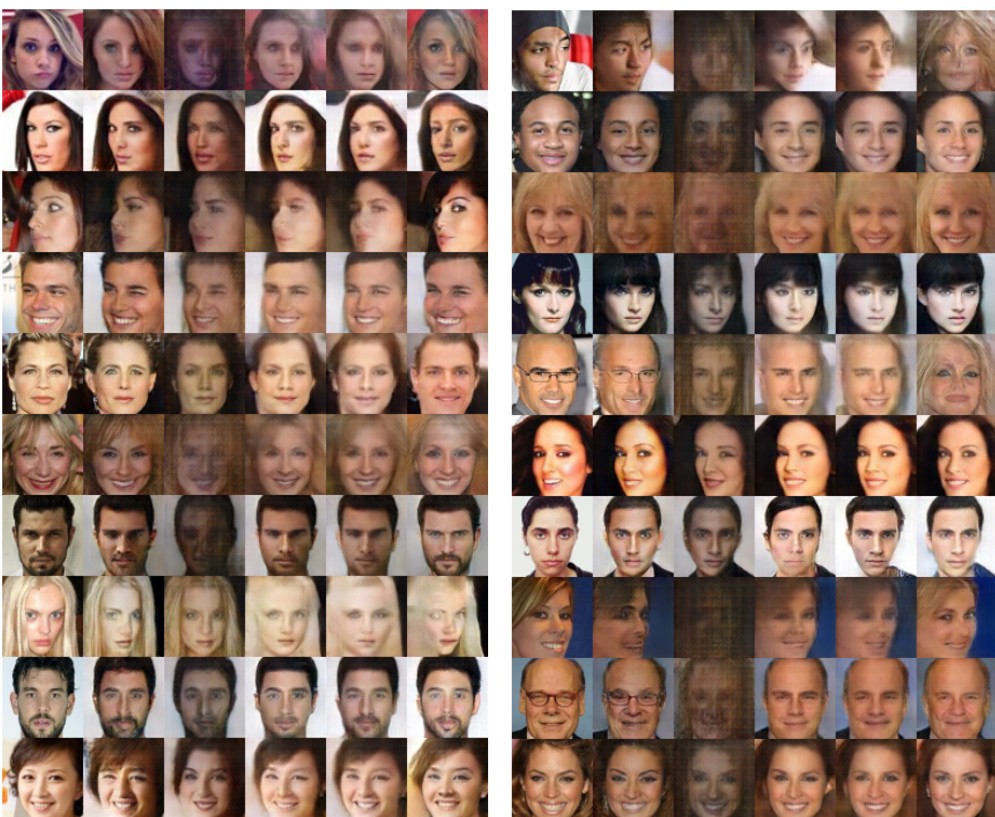

Figure 6: Qualitative comparison of reconstruction by each method; Test, Ours, Naïve encoder, iGAN, Hybrid+Ours, and VGG16.

In the section 4, we quantitatively and qualitatively evaluate the proposed inference mapping compared with three different inference methods and confirm recycling discriminator is suitable for the feature extractor. Furthermore, we analyze the reason for the gap between quantitative results and qualitative results. Although our quantitative results are somewhat worse than those of hybrid methods, we believe that our inference mapping (i.e., adopting the inference mapping through optimizing on the latent space) is still the better choice. It is because our inference mapping preserves the desirable properties of GANs, which generate sharp images with various semantic attributes. We believe that reproducing the semantic attributes could be interpreted as how well GANs understand the images. In this regard, because the hybrid method tends to choose average-like images even omitting semantic attributes, it is not appropriate to adopt the hybrid method for building the GAN evaluation metric. Also, by recycling the discriminator, the quality of discriminator directly affects the accuracy of inference mapping. Thus, our method is suitable for evaluating the entire GAN framework; we utilize both the generator and the discriminator for evaluating the GAN model.

