# OpenReview forum: "Recycling the discriminator for improving the inference mapping of GAN"
_ICLR.cc/2019/Conference_

### Official Review · AnonReviewer1 · 2018-11-01
**a novel approach to GAN inference mapping**

**Rating:** 7
**Confidence:** 4

**Review:**

This paper describes a novel method to provide inference mapping for GAN networks. The idea is to reuse the discriminator network's feature vector (output of layer before last) and learn a direct mapping to the GAN's latent space. This can be done very efficiently since the dimensionality of both layers are relatively small. Also, the mapping does not interfere with the learning process of the GAN itself and thus can be applied on top of any GAN method without affecting its performance.

Inference mapping is useful in the GAN context for several reasons that are well described in the paper. First it allows to more efficiently generate "edited" images as the mapping provides a good starting point in the latent space. Second it provides a sound way to evaluate GAN's performance as the reconstruction of a given image through the inference mapping and the generator provides auto-encoder-like capabilities. Comparison of GAN models have been difficult due to a lack of adequate evaluation technique. This paper proposes a novel evaluation scheme that is both fair and technically simple.

In the experimental part, the authors first compare their approach to the 'naive encoder' approach where the last layer of the discriminator is removed after training, a feature layer of the size of the encoder's latent space is added, and the rest of the discriminator's layers are frozen. The proposed approach outperforms the naive encoder approach on the CelebA dataset. The second set of experiments investigates reconstruction accuracy of various GAN models. Figure 2 shows reconstructed images for 7 GANs and 36 examples from 3 datasets. Unfortunately, no subjective comparison can be attempted since the examples are different for each GAN. In Figure 3, editing in performed on the CelebA dataset, but again, subjective comparison among the GAN's is precluded by the fact that different examples are chosen. This oversight does not affect the paper's relevance, since those comparison would be purely subjective, however it would add some visual interpretation to the quantitative comparison given in table 1. I also wish the authors would have provided the inception score for FashMNIST and CelebA and also provide the more recent FID (Frechet Inception Distance). Inception scores are trained on ImageNET and are too commonly applied to CIFAR-10 and CelebA. It would be good to compare them against the proposed method on those datasets to show that there are not good for datasets other than those on which they were trained.

The article is technically sound. The citations are adequate. The English is fine with some extraneous articles being the only issue. The article lacks a graphic for the architecture of the system and many of the figures are too small to interpret when printed out. Also there's a typo on table 1. where the inception score for WGAN-GP on CelebA should be 6.869 and not 0.6869.

Overall, I find this paper provides a simple, novel significant method for evaluating GAN models and making better use of their latent space arithmetic editing capabilities. Due to the algorithm's simplicity, most of the paper is devoted to experiments and discussions.

---

> ### Author Response · Authors · 2018-11-16
> **Author Response to Reviewer 1**
>
> Thanks for your comments,
> We thank the Reviewer1 for constructive feedback. Reviewer1 suggests the comparison among the results of various unidirectional GANs to strengthen the experimental evaluation. We agree that the comparison mentioned by Reviewer1 helps improving the quality of the paper. To reflect this comment, we now revise our manuscript to supplement the qualitative comparison among unidirectional GANs.
> Furthermore, Reviewer1 suggests that you should provide the inception score or MS-SSIM of other datasets. It would be helpful to show that the existing metrics are meaningful only on the specific datasets. As we mentioned on section 3.2, MS-SSIM scores are almost zero for CIFAR10 or Fashion MNIST, which have multiple classes. Meanwhile, the inception score utilizes the classifier, thus it is not appropriate to apply onto the single class dataset (i.e., CelebA). As a result, the scores from those cases would present meaningless scores. Due to limited space, we did not include them in our main manuscript.
> We hope our additional results can answer your comments.

---

### Official Review · AnonReviewer2 · 2018-11-02
**Insufficient experimental validation and marginal novelty**

**Rating:** 3
**Confidence:** 4

**Review:**

The paper proposes using the GAN discriminator for inference mapping, mapping an image to a latent code that would be used to generate the image by the encoder, based on the argument that the discriminator can be used as a powerful feature extractor because it has seen both real and fake data during training. The paper compares the proposed approach to several approaches that train an inference model together with a generator.

While the paper compares its approach to several baselines, they are not the most relevant ones. In fact, the most relevant baseline is not cited and compared. As a result, the novelty of the paper is not justified. Specifically, the baselines the paper compare to are mostly methods that jointly learn an inference model and a generation model, while the proposed approach first learns a generation model and then fits an inference model (it is referred to as the connection network in the paper). In this regard, the paper should compare its approach to methods that first learns a generation model and then learns an inference model. The iGAN work by Zhu et. al. ECCV 2016 is arguably most relevant approach. Especially, they also use the discriminator architecture for the inverse mapping. Unfortunately, the work is neither cited nor compared.

In addition, pretrained networks such as VGG and ResNet have been known to be powerful feature extractor. It would be ideal the paper can compare the proposed approach to that using VGG and ResNet for finding the z for a given image.

Finally, the paper seems to lack of comprehensive knowledge on how the inference mapping has been investigated in the GAN literature. For example,  the statement that "BEGAN (Berthelot et al., 2017) made the first attempt to solve the inverse mapping from x to z using the non-convex optimization" in the introduction section is incorrect. The scheme is used in at least two 2016 papers (Liu and Tuzel NIPS 2016 and Zhu et. al. ECCV 2016).

---

> ### Author Response · Authors · 2018-11-14
> **Author Response to Reviewer 2_2**
>
> Our algorithm successfully synthesizes the attributes in various faces, unlike the na{\"i}ve encoder. As reported in iGAN, we confirm that adopting the non-convex optimization for inference mapping significantly enhances the quantitative score (i.e., SSIM and PSNR). It is because the non-convex optimization directly minimizes the pixel-wise difference between test images and reconstructed images; the goal of the non-convex optimization is nearly equivalent to the goal of PSNR. Hence, the hybrid method improves the PSNR of any baseline encoder mapping. When we replace the na{\"i}v encoder with the proposed inference mapping, its quantitative results are better than iGAN. It is because our inference mapping predicts more accurate initial latent vector.
>
> However, these quantitative results do not exactly match with qualitative results. The quantitative results demonstrate that hybrid inference mapping is the most effective among all others. Meanwhile, the qualitative results from the hybrid methods are generally blur or have missing important components (e.g., eye glasses, mustache, gender, wrinkles, detailed hair lines, etc.). Because the hybrid inference mapping optimizes the inference mapping in the image domain (i.e., minimizing the pixel-wise difference), the inference network finally chooses the latent vector corresponding to an average-like image. Note that there exists average-like faces among many possible faces. We conjecture that, although the generator can produce sharp images, the hybrid inference mapping strategically selects average-like faces to reduce its loss function. Meanwhile, our method (also VGG16 based inference mapping) optimizes the inference mapping in the latent domain. Thus, our inference results are sharp and better preserve semantically important attributes. From examples shown in Fig ~\ref{figure04} and Appendix Fig  \ref{appendix fig}, pixel-wise loss based methods  (i.e., iGAN and Hybrid+ours) fail to capture glasses, but latent vector loss based methods  (i.e., ours and VGG16) reproduce the glasses. In fact, for the same reason, VEEGAN chooses to minimize a reconstruction loss on the latent vector to solve mode collapse.
> By replacing the discriminator as feature extractor by the pre-trained VGG16 network, we observe that its inference results are also as sharp and realistic as our results. However, considering the semantic similarity between the original and reconstructed image, our inference mapping can restore unique attributes (e.g., mustache, race, age, etc.) better than the VGG16 based inference mapping. Moreover, utilizing the pre-trained VGG16 require additional memory overhead while our method does not. In terms of network capacity, VGG16 has the much deeper network than the discriminator. Thus, we conclude that the proposed inference mapping is more efficient than the VGG16 based inference mapping. From these results, we confirm that recycling discriminator as a feature extractor is effective for improving inference accuracy and reducing the computational complexity.
>
> In conclusion, we adopt the inference mapping through optimizing on the latent space because this preserves the properties of GANs that generate sharp images and semantic attributes. Moreover, since preserving the semantic attributes could be interpreted as how well GAN understand the images, the non-convex optimization that prefers average images even omitting semantic attributes is not appropriate for suggesting the GAN evaluation metric. In the same context, since recycling the discriminator directly affect the inference mapping accuracy, our method is suitable for evaluating whole GAN framework (i.e., both the generator and the discriminator).

---

> ### Author Response · Authors · 2018-11-14
> **Author Response to Reviewer 2_1**
>
> Thanks for your comments,
> We appreciate the constructive feedback from Reviewer 2. As Reviewer 2 pointed out, we missed the comparison with the relevant existing work (iGAN) which also adopts independent inference mapping method to the GAN training; our experimental evaluation focused on the limitation of the bidirectional generative models that trains generation and inference mapping together. We agree that the comparison with iGAN and the ablation study using the pre-trained feature extractor would improve the quality of our paper. To reflect this valuable feedback, we conduct the experiment as follows. Please note that all changes are reflected in our revision (page 6-7, 13).
>
> iGAN compares three different inference prediction methods. 1) The first method is a direct inference mapping through naïve encoders. 2) The second method is to adopt the non-convex optimization, which minimizes the pixel wise difference between the original image and the image generated from the estimated latent vector. 3) Finally, their proposal is a hybrid method that carries out encoder mapping followed by the non-convex optimization. Similarly, we compare the proposed method with 1) naïve encoder mapping, 2) iGAN (naïve encoder followed by optimization) and 3) hybrid method using the proposed method (discriminator with CN followed by optimization). Furthermore, to investigate the capability of the discriminator as the feature extractor, we directly compare our inference mapping (discriminator with CN network) with the VGG-16 based inference mapping (pre-trained VGG with CN network). Note that the pre-trained VGG16 network is trained on ImageNet 1k. The quantitative evaluation is summarized as follows. Please also see qualitative results on our revision.
>
>
>                 Ours	naïve	iGAN	Hybrid+Ours	VGG16
> SSIM	0.5214	0.4872	0.5624	0.5710		0.5199
> PSNR	16.85	15.28	18.21	18.52		16.81

---

### Official Review · AnonReviewer3 · 2018-11-06
**Missing important connections to existing works**

**Rating:** 3
**Confidence:** 5

**Review:**

Paper Summary:
This paper proposes to reconstruct the generated images to the their corresponding latent code. As claimed, the goal is to improve the accuracy and efficiency of inference mapping better than other inference mapping techniques, while maintaining their generation quality.
 Instead of using an independent encoder, the authors propose to share the encoder parameters with the discriminator: a Connection Network (CN) is built on top of the features extracted by the discriminator. The weight-sharing machisme shows better performance in Figure 1.
The proposed method has two benefits: : a) manipulating the image by disentangling the latent space and b) suggesting a new metric for assessing the GAN model by measuring reconstruction errors of real data.

General Comments:
In term of algorithm, the paper essentially adds the conscontruction term (CN) to the standard GAN loss, and partially shares the weights of the “encoder” and discriminator. However, it is almost identical to the existing works, which are NOT cited, and the connections are not discussed.

Connection to InfoGAN: To relate the generated images to the latent code,  the proposed method employs the reconstruction loss, InfoGAN employs the mutual information. Note that reconstruction loss = negative log likelihood, and effectively is equivalent to Mutual Information and Conditional Entropy in the case. Please see the discussion in Lemma 3 and Appendix A of [3] for detailed discussion.  Further, InfoGAN has proposed to to sharing weights of the encoder and discriminator, exactly the same with this submission. The claimed advantage is to disentangle the latent space. It is not surprise at all, once the authors see the connection to InfoGAN, which was originally proposed to disentangle the latent codes.

Connection to CycleGAN: CycleGAN consists of four losses: two reconstruction losses and two standard GAN losses. As shown in Section 4 of [3] “Connecting ALI and CycleGAN”, one reconstruction loss and  one standard GAN loss is sufficient to achieve CycleGAN’s objective, the other two losses would only help to accelerate. In another word, the proposed method is exactly half of the CycleGAN losses.

The author mention in Abstract that “the bidirectional generative models introduce an encoder to establish the inverse path of the generation process. Unfortunately, their inference mapping does not accurately predict the latent vector from the data because the imperfect generator rather interferes the encoder training.” This is the non-identifiable issue of ALI/BiGAN discovered in [3]. Please clarify.

The proposed method should compare with [1] and [2] in great detail, to demonstrate its own advantages. Given the missing literature, the current experimental comparisons seem not that meaningful, because the baseline methods are not really the competitors.

One interesting contribution of the submission is to consider the reconstruction errors to measure the quality of GANs. To my best knowledge, it is original.


References:

[1] InfoGAN: Interpretable Representation Learning by Information Maximizing Generative Adversarial Nets, NIPS 2016
[2] Unpaired Image-to-Image Translation using Cycle-Consistent Adversarial Networks, ICCV 2017
[3] ALICE: Towards Understanding Adversarial Learning for Joint Distribution Matching, NIPS 2017

---

> ### Author Response · Authors · 2018-11-10
> **Author Response to Reviewer 3**
>
> Thanks for your comments,
> First of all, we would like to solve important misunderstanding about our methodology. Especially, the  major difference of our work compared to three techniques mentioned by Reviewer 3 is summarized as follows. Unlike the existing techniques (i.e., InfoGAN, CycleGAN and ALICE), our inference mapping using the connection network is completely independent of both generator updates and discriminator updates. We would like to stress that this is why we could maintain the quality of generation in the baseline GAN model; other inference mapping techniques influence the generation quality.
>
> Our key idea of inference model is to reuse the discriminator network as feature extractor and learn a direct mapping from the feature vector to the GAN's latent space, as mentioned by Reviewer 1. Because we utilize the well-educated discriminator to extract the meaningful features, the direct mapping is learned after the training of both generator and discriminator end. On the other hand, infoGAN, CycleGAN, and ALICE intend to design the inference mapping that controls the generation process; the above three techniques all affect the generator updates for their own purpose. Again, we emphasize that the reconstruction loss of the connection network is different from the conditional entropy of infoGAN and the cycle consistency of CycleGAN and ALICE in two aspects. First, our model does not affect both the generator update and the discriminator update. Secondly, our goal is to build the inference mapping without affecting the baseline GAN performance. Meanwhile, three techniques develop new GAN models for learning interpretable representation (infoGAN), unsupervised domain transfer (CycleGAN), or alleviating the mode collapse (ALICE).
> We decide to separate the generation mapping (from z to x) and the inference mapping (from x to z) because of the convergence issue reported in bidirectional GANs. For example, ALICE reported in appendix E.3, “As a trade-off between theoretical optimum and practical convergence, we employ feature matching, and thus our results exhibit a slight blurriness characteristic”. We believe that this convergence issue is caused by the error propagation from both generator and encoder. Furthermore, their inception score is 6.015, which is slightly worse than the average inception of unidirectional GAN, 6.5. This experimental result demonstrates that existing bidirectional methods (either using cycle consistency or joint distribution matching) have shown the limited generation performance. Although we did not compare the performance to ALICE directly in the paper, the experimental results of ALI/BiGAN could represent the limitation of bidirectional GANs trained for joint distribution matching.
>
> We agree that the reconstruction loss could be interpreted as the negative log likelihood, and this is equivalent to mutual information and conditional entropy. However, such a scheme is applicable for techniques that handle the joint distribution (both generation and inference or both generation and latent code) matching. Meanwhile, our model disconnects the inference mapping from generation process.
>
> We now revise our manuscript to clarify those of difference of ours and three existing techniques. We hope our explanations can answer your concerns.

---

### Meta-Review · Area_Chair1 · 2018-12-14
**Marginal novelty; the advantage over existing methods is not convincing enough**

**Confidence:** 5
**Recommendation:** Reject

**Metareview:**

The paper presents a method to learn inference mapping for GANs by reusing the learned discriminator's features and fitting a model over these features to reconstruct the original latent code z. R1 pointed out the connection to InfoGAN which the authors have addressed. R2 is concerned about limited novelty of the proposed method, which the AC agrees with, and lack of comparison to a related iGAN work by Zhu et al. (2016). The authors have provided the comparison in the revised version but the proposed method seems to be worse than iGAN in terms of the metrics used (PSNR and SSIM), though more efficient. The benefits of using the proposed metrics for evaluating GAN quality are also not established well, particularly in the context of other recent metrics such as FID and GILBO.